# Seroprevalence of *Toxoplasma gondii*, *Neospora caninum* and *Trichinella* spp. in Pigs from Cairo, Egypt

**DOI:** 10.3390/vetsci10120675

**Published:** 2023-11-27

**Authors:** Ragab M. Fereig, El-Sayed El-Alfy, Hanan H. Abdelbaky, Nour H. Abdel-Hamid, Amira M. Mazeed, Ahmed M. S. Menshawy, Mohamed A. Kelany, Mohamed El-Diasty, Bader S. Alawfi, Caroline F. Frey

**Affiliations:** 1Division of Internal Medicine, Department of Animal Medicine, Faculty of Veterinary Medicine, South Valley University, Qena 83523, Egypt; 2Department of Parasitology, Faculty of Veterinary Medicine, Mansoura University, Mansoura 35516, Egypt; sydnabil@mans.edu.eg; 3Doctor of Veterinary Sciences, Veterinary Clinic, Veterinary Directorate, Qena 83523, Egypt; hananragabegypt@gmail.com; 4Brucellosis Research Department, Agricultural Research Center, Animal Health Research Institute, Cairo 12618, Egypt; nour78_78@ahri.gov.eg; 5Department of Infectious Disease, Faculty of Veterinary Medicine, Arish University, Arish City 45511, North Sinai, Egypt; amera.mohamed@vet.aru.edu.eg; 6Department of Veterinary Medicine, Faculty of Veterinary Medicine, Beni-Suef University, Beni-Suef 62511, Egypt; ahmed.elmenshawy@vet.bsu.edu.eg; 7Department of Microbiology, The Central Laboratory of Residual Analysis of Pesticides and Heavy Metals in Foods, Agricultural Research Center, Dokki, Giza 12618, Egypt; mohamed_kelany@hotmail.com; 8Agricultural Research Center (ARC), Animal Health Research Institute-Mansoura Provincial Lab. (AHRI-Mansoura), Giza 12618, Egypt; m.mesbah@ahri.gov.eg; 9Department of Clinical Laboratory Sciences, College of Applied Medical Sciences, Taibah University, Madinah 42353, Saudi Arabia; bawfi@taibahu.edu.sa; 10Institute of Parasitology, Department of Infectious Diseases and Pathobiology, Vetsuisse-Faculty, University of Bern, Länggassstrasse 122, CH-3012 Bern, Switzerland

**Keywords:** pigs, Egypt, toxoplasmosis, neosporosis, trichinellosis, ELISA

## Abstract

**Simple Summary:**

*Toxoplasma gondii* and *Trichinella* species are parasites of major public health importance because they cause severe clinical consequences in infected humans. In addition, infections of *Neospora caninum* in livestock are reported to cause substantial losses to the economy because of adverse effects on animal production and reproduction. In Egypt, the breeding of pigs has particular characteristics regarding the targeted consumers (tourists and Christians), breeding places (Cairo and Giza), and licensed abattoirs (only El-Bassatin in Cairo). Herein, we sought to provide a comprehensive investigation of the major parasites of pigs, including *T. gondii*, *Trichinell*a species, and *N. caninum*, a parasite of high priority in ruminants. This study revealed variable seropositive rates against all tested parasites in pigs from Cairo, Egypt. Seropositivity was the highest for *T. gondii* (45.8%), followed by *N. caninum* (28.0%), mixed infection by both parasites (18.7%), and *Trichinella* spp. (1.2%). We also found the location was a predisposing factor for seropositivity for *T. gondii*, while location and sex were identified as predisposing factors for *N. caninum* seropositivity in pigs. The provided information in this study presents valuable information on the seroprevalence of *T. gondii* and *Trichinella*, and novel information on *N. caninum* existence among pigs in Cairo, Egypt.

**Abstract:**

Pork production is a niche economy in Egypt, and pigs are typically raised as backyard animals with no sanitary control, potentially exposing them to various pathogens. Commercially available ELISAs were used to detect specific antibodies to the food-borne zoonotic parasites *Toxoplasma gondii* and *Trichinella* spp., as well as to *Neospora caninum*, in serum samples of pigs slaughtered at Egypt’s only licensed pig abattoir, the El-Bassatin abattoir in Cairo. Among the tested sera (n = 332), seroreactivity for *T. gondii* was 45.8% (95% confidence interval: 40.4–51.3), *N. caninum* was 28.0% (95% CI: 23.3–33.2), and *Trichinella* spp. was 1.2% (95% CI: 0.4–3.3). Mixed infection was only detected for *T. gondii* and *N. caninum* (18.7%; 95% CI: 14.7–23.4). The seroprevalence of *T. gondii* was significantly higher (*p* = 0.0003) in animals collected from southern Cairo (15 May city slum) than in eastern Cairo (Ezbet El Nakhl slum). Seroprevalence for *N. caninum* was higher in western (Manshiyat Naser slum; *p* = 0.0003) and southern Cairo (15 May city slum; *p* = 0.0003) than in that of eastern Cairo (Ezbet El Nakhl slum; *p* = 0.0003). Moreover, female pigs exhibited a higher rate of *N. caninum* antibodies than male ones (*p* < 0.0001). This study provides the first seroprevalence data for *N. caninum* in pigs in Egypt, and updates the prevalence of the zoonotic parasites *Trichinella* spp. and *T. gondii*.

## 1. Introduction

A lack of food security is one of the issues plaguing the world’s rapidly growing human population, and the global prevalence of malnutrition increased dramatically between 2019 and 2020, owing mostly to the COVID-19 pandemic [1]. The population in Egypt is growing fast; the estimated population in 2021 was 109.3 million, with a population change rate of 1.59% [2]. Egypt is primarily a Muslim country; however, it has lately been estimated that approximately 10% of the Egyptian population are Christians [3]. The swine/pigs stock number in the country was estimated as 11,000 head by the year 2021 [4]. Pigs in Egypt are typically reared as backyard animals and fed on human garbage [5,6]. Consequently, it is critical to investigate pigs for parasites that are either of zoonotic or veterinary importance.

With a global distribution, *Toxoplasma gondii* infects nearly all warm-blooded animals, including humans, livestock, and marine mammals [7,8]. The definitive hosts are felids, which produce and shed environmentally-resistant oocysts [8,9]. Pigs, like humans and other animals, become infected with *T. gondii* after ingesting sporulated oocysts shed by cats or tissue cysts in the meat of other intermediate hosts [10]. Natural *T. gondii* infection in pigs is a public health issue, as transmission to humans by consumption of raw or undercooked pork meat or offal can occur [11]. Infections in pigs are usually asymptomatic or with non-pathognomonic symptoms, and severe clinical toxoplasmosis is rare [11]. Human primary infection with *T. gondii* normally manifests as mild flu-like symptoms in immune-competent hosts; however, it can induce life-threatening infections in immune-compromised persons. Furthermore, *T. gondii* transplacental transmission during pregnancy might result in significant complications such as abortion, stillbirth, or congenital anomalies [8].

The pooled global *T. gondii* seroprevalence in pigs (148,092 pigs from 47 countries) was estimated to be 19% based on different serological methods, among which enzyme-linked immunosorbent assays (ELISA) were used most frequently [12]. In Egypt, the reported seroprevalence in pigs varies between 14% and 80.4% [13,14,15,16,17,18,19,20,21]. Additionally, viable *T. gondii* was isolated from the heart of a healthy seropositive pig in Egypt. Many cases of human toxoplasmosis have been reported in Egypt and reviewed previously. The reported seroprevalence of *T. gondii* in the Egyptian human population ranges from 2.5% to 67.4%, and toxoplasmosis is commonly regarded by Egyptian physicians as a cause of abortions and complications in pregnant women; nonetheless, the published studies are not well-structured, and there is a lack of definitive diagnoses [20,21,22].

Neosporosis is a disease caused by the heteroxenous cyst-forming parasite *Neospora caninum*, an intracellular apicomplexan protozoan that infects a wide range of domestic and wild animals [23]. The definitive hosts of *N. caninum* are domestic and wild dogs, coyotes, and grey wolves [24,25]. Cattle are the most common intermediate hosts of *N. caninum*; however, sheep, water buffaloes, bison, and white-tailed deer have also been identified as intermediate hosts [26]. Abortions and newborn mortality are serious complications of neosporosis in livestock, particularly in cattle [23,24].

Anti-*N. caninum* antibodies have been reported in domestic and feral pigs from various countries [24], and, more recently, parasite DNA has been detected in the brains of domestic pigs in China [27], indicating natural infection. However, the pathogenesis of neosporosis and its consequences in pigs remains unclear [28,29]. The detection of *N. caninum* infection in live animals is based mainly on the identification of anti-*N. caninum* antibodies using a variety of serological tests, the most frequently used of which is ELISA [30,31]. In Egypt, *N. caninum* infections have been detected in domestic animals such as cattle, buffaloes, and camels, but there have been no reports in pigs.

*Trichinella* nematodes are zoonotic parasites with a global distribution [32]. Pork is the most common source of human trichinellosis, which can cause clinical symptoms such as myalgia, diarrhea, fever, facial edema, and headaches, and may even be fatal [33]. *Trichinella spiralis* is the only species reported in Egypt so far [32], and it has been identified in pigs [34,35], rodents [36], and dogs [34,37]. In Egypt, the reported infection rate in slaughtered pigs and pork products ranges from 0% to 6% [19,35,38,39,40,41]. Only one old report on the seroprevalence of *Trichinella* spp. in pigs in Egypt (35.6%) was found in a literature search, and it used an immunofluorescent antibody test (IFAT) [42]. Nevertheless, ELISA is the recommended assay for epidemiologic serosurveillance of pigs for *Trichinella* infection [43]. Human trichinellosis linked to the consumption of pork has recently been reported from Egypt based on ELISA, and the seroprevalence rate of *T. spiralis* (IgG) was 10% (9/90) [35] and 60.9% (56/92) [40] in the examined humans.

Based on the above-mentioned facts, there have been a few reports in Egypt on the prevalence of *T. gondii* and *Trichinella* spp. in pigs, but none on *N. caninum*. We therefore opted to update the information on the seroprevalence of *T. gondii* and *Trichinella* spp., as well as to provide baseline data for the seroprevalence of *N. caninum* infections, in pigs in Egypt. Our presented data will serve the public health and veterinary sectors in minimizing the health and economic hazards of these parasites in Egypt.

## 2. Materials and Methods

### 2.1. Description of the Animals and Regions of the Study

Blood samples were collected from male and female pigs (n = 332) during slaughtering in the El-Bassatin abattoir in the Cairo governorate. All pigs were raised in backyard pens containing 50 to 100 pigs in the slums of the Cairo governorate by garbage collectors as one of their main sources of income. These rearing pens and slums lack many procedures ensuring sanitary conditions or biosecurity and pigs are fed mainly on garbage. This condition allows for high rates of contact with free-roaming animals such as dogs, cats, rodents, and birds, increasing the incidence of infection with many biohazardous agents. In addition, other livestock animals, especially sheep, are also reared with these pigs. The majority of pigs are reared for fattening and are slaughtered at the age of 5 months. They are typically sold to pig merchants who collect animals from different small-scale farmers and ship them to the abattoir. Pigs used in this study originated from the following slums: Manshiyat Naser, Cairo’s largest pig farming slum in the western region (30.0362°N, 31.2783°E) (n = 108; female 75, male 33); Ezbet El Nakhl (30.1393°N, 31.3244°E) (n = 184; female 125, male 59) in the eastern Cairo region; and 15 May city (29.8579°N, 31.3885°E) (n = 40; all of them male) in the southern Cairo region (Figure 1). The pigs from various herds were held together in abattoir pens before slaughtering without identification.

Blood was collected in plain tubes without anticoagulants directly from the slaughtered animals during bleeding according to the El-Bassatin abattoir regulations for pig slaughtering and pork processing. All samples were labeled and transferred on ice to the Reference Laboratory at the Animal Health Research Institute (AHRI), Giza. Upon arrival at the laboratory, blood samples were kept overnight for serum separation and then centrifuged at 900× *g* for 10 min. Serum samples were aliquoted and kept at −20 °C until testing. Aliquots from these samples were sent on ice to the Faculty of Veterinary Medicine, South Valley University, Qena, for ELISA testing.

### 2.2. ELISAs for Screening Antibodies to T. gondii, N. caninum, and Trichinella Species in Pigs

Regarding *T. gondii*, the serum samples were tested using an indirect multi-species ELISA for toxoplasmosis (ID.vet, Grabels, France) according to the manufacturer’s instructions. Serum samples and controls were diluted 1:10. The optical density (OD) obtained was used to determine the percentage of sample (*S*) to positive (*P*) ratio (S/P %) for each of the test samples using the following formula: S/P (%) = (OD sample – OD negative control)/(OD positive control − OD negative control) × 100. Samples with an S/P% less than 40% were regarded as negative, an S/P% between 40% and 50% was regarded as doubtful, and the test was considered positive if the S/P% was more than 50%.

For *N. caninum*, serum samples were analyzed using a competitive multi-species ELISA for neosporosis (ID.vet, Grabels, France). Serum samples and controls were diluted 1:2. The ODs obtained were used to calculate the percentage of sample (*S*) to negative (*N*) ratio (S/N%) for each of the test samples according to the following formula S/N (%) = OD sample/OD negative control × 100. Samples with an S/N% more than 60% were regarded as negative, an S/N% between 50% and 60% was regarded as doubtful, and the test was considered positive if the S/N% was less than 50%.

Serum samples were analyzed for *Trichinella* species using an indirect multi-species ELISA for trichinellosis (ID.vet, Grabels, France). Serum samples and controls were diluted 1:20. The ODs obtained were used to determine the percentage of sample (*S*) to positive (*P*) ratio (S/P %) for each of the test samples using the following formula: S/P (%) = (OD sample − OD negative control)/(OD positive control − OD negative control) × 100. Samples with an S/P% less than 50% were regarded as negative, an S/P% between 50% and 60% was regarded as doubtful, and the test was considered positive if the S/P% was greater than or equal to 60%.

The ODs of all ELISA results were read at 450 nm using an Infinite^®^ F50/Robotic ELISA reader (Tecan Group Ltd., Männedorf, Switzerland). See Table 1 for additional information on the ELISAs used in this study.

### 2.3. Statistical Analysis

The significance of the differences in the prevalence rates and risk factors were analyzed using Fisher’s exact test, 95% confidence intervals (95% CI) (including a continuity correction), and odds ratios (OR) using an online statistical website www.vassarstats.net (accessed on 22 July 2023). From this website, calculator 3 was selected from the clinical research calculators. Then, in squares of data entry, the numbers of negative (absent) and positive (present) samples from group 1 (regarded as the reference group) were added. Similarly in the lower squares, the numbers of negative and positive samples were added. In these tests, we used the two-tailed *p* values of Fisher’s exact test. *p*-values and odds ratios were confirmed with GraphPad Prism version 5 (GraphPad Software Inc., La Jolla, CA, USA) [44,45].

## 3. Results

The overall seroprevalence in the studied pigs for *T. gondii* was 45.8% (152/332; 95% CI: 40.4–51.3), *N. caninum* was 28.0% (93/332; 95% CI: 23.3–33.2), and *Trichinella* spp. was 1.2% (4/332; 95% CI: 0.4–3.3). Mixed infections were only detected for *T. gondii* and *N. caninum* at 18.7% (62/332; 95% CI: 14.7–23.4) (Table 2).

Risk factor analysis was conducted to assess the influence of sex and breeding location on the seroprevalence of *T. gondii* and *N. caninum*, but not for *Trichinella* because of the limited number of seropositive samples. Breeding site was identified as the only predisposing factor for *T. gondii* infection. The seroprevalence of *T. gondii* in pigs reared in 15 May city, southern Cairo (67.5%) set as the reference group was higher than that in Ezbet El Nakhl, eastern Cairo (35.3%) (OR = 3.8, *p* = 0.0003), but not significantly higher than in Manshiyat Naser, western Cairo (55.6%) (OR = 0.6, *p* = 0.259). On the other hand, sex had no influence on the *T. gondii* seroprevalence (Table 3).

For *N. caninum* antibody levels, both sex and breeding site were considered risk factors. The seropositive rate was significantly higher in female pigs (38%) set as a reference than in male pigs (12.9%, OR = 4.1, *p* ≤ 0.0001). Also, the seroprevalence of *N. caninu*m in pigs located in Ezbet El Nakhl (13%) set as the reference group was significantly lower than that recorded in Manshiyat Naser (53.7%, OR = 7.7, *p* ≤ 0.0001) and 15 May city (27.5%, OR = 2.5, *p* ≤ 0.031) (Table 4).

## 4. Discussion

Despite the presence of a considerable pig population and pig consumers in Egypt, there has been relatively little effort to conduct parasite epidemiological studies. Herein, we targeted pigs as sentinel animals for infection with three parasites: *T. gondii*, *N. caninum*, and *Trichinella* species. All animals were admitted to El-Bassatin abattoir, Cairo, the only licensed place for pig slaughtering in Egypt. The highest seroprevalence was recorded for *T. gondii* (45.8%), followed by *N. caninum* (28.0%) and *T. gondii* and *N. caninum* co-infection (18.7%), and the lowest was for *Trichinella* spp. (1.2%).

A variety of techniques and antigens are available for the detection of distinct immunoglobulin isotypes against *T. gondii* [46,47], among which ELISA has been the most frequently used method in the past century [48]. In Egypt, few serological studies have been published for *T. gondii* infections in pigs, and prevalence rates between 14% and 80.4% have been reported [13,14,15,16,17,18,19,20,21]. These studies were conducted over a large time span (1972–2015), and a multitude of test types were used, such as indirect fluorescent antibody test (IFAT) [13,16], indirect hemagglutination assay (IHA) [19,20], dye test (DT) [14], modified agglutination test (MAT) [15], and ELISA [17,18]. In the present study, we opted for commercially available ELISAs, which have the advantage of good repeatability, robustness, and reportedly very good sensitivity and specificity. The detected seroprevalence of 45.8% for *T. gondii* in the assessed backyard pigs indicates a potential risk to the health of pig fatteners, abattoir workers, and consumers. A similar finding was published by Barakat et al. [16], who reported a seroprevalence for *T. gondii* of 74.7% in pigs reared entirely on waste feed in Cairo, and a seroprevalence of 37.7% (48 of 127) in the workers at these pig farms.

*Neospora caninum* infection in Egyptian animals is underestimated, with a few seroprevalence reports in cattle, buffaloes, sheep, goats, camels, dogs, and cats, as well as limited information on clinical neosporosis. In our previous reports using the same ELISA as in the current study, we demonstrated seroprevalence rates for *N. caninum* of 3.9% in camels [44], 15.5% and 5% in sheep and goats, respectively [45], 5.8% in dogs, 3.4% in cats [49], and 24.6% in cattle [50]. Nonetheless, serum samples from aborted cows revealed *N. caninum* infections in addition to *Coxiella burnetii* [51] or Bovine Viral Diarrhoea Virus [52]. Furthermore, the number of *N. caninum* seropositive camels increased among females with an abortion history [53]. On the other hand, the pathogenesis of neosporosis and its consequences in pigs remains unclear [28,29]. *Neospora caninum* experimental infection in pigs can be transferred through the placenta and produce reproductive problems such as mummified fetuses, particularly in the first and second gestational thirds [28].

To the best of our knowledge, the current study is the first serological study of *N. caninum* infection in pigs in Egypt. Thus, we compared our recorded prevalence (28%) with the rates in other animals in different regions of Egypt. Although *N. caninum* might not be a serious parasite for the pig industry, its detection in pigs might constitute a major risk for ruminants and other animals because of multiple species being raised in the same place. The seroprevalence in pigs was most similar to rates reported in cattle (24.6% [50], 29% [51], and 30.2% [52]) and higher than the rates reported in all other animal species, with the exception of one study in buffaloes (Table 5). Furthermore, in comparison with the seroprevalence rate of *N. caninum* in domestic pigs from other countries, our infection rate seemed higher. For example, a rate of 3% was reported in the Czech Republic [54]; 1.9% in Hunan province, China [27]; 5.2% in Brazil [55]; and 6.7% in Italy [56] for *N. caninum* infections in pigs. Only one study found a higher seroprevalence rate (36.4%) in Qinghai-Tibetan plateau, China [57].

Overall, these findings can best be explained by the sanitary conditions in which the pigs were raised. They were garbage-fed backyard pigs and fattened as an additional income by inhabitants of the slums of Cairo. Both parasites can be transmitted through the feeding of uncooked infected meat from other species or via oocysts shed by cats for *T. gondii* or by dogs for *N. caninum* [8,23]. Slums typically harbor a large population of stray cats, dogs, and other animals that may serve as sources of infection for the pigs.

Little is known about the oocyst-shedding rates of cats and dogs in Egypt. The presence of *T. gondii*-like oocysts in the feces of cats in Egypt may have been overestimated, where 1432 cats were examined, revealing a pooled prevalence of 11.9% [64]. *Neospora caninum*-like oocysts were found in the feces of one of 78 dogs tested in a single study (Dakahlia governorate), and it is conceivable that certain small-sized oocysts detected in prior studies from Egypt are *N. caninum* [65]. Furthermore, *N. caninum*-like oocysts were found in the feces of five of nine puppies fed the placentas and brains of nine aborted bovine fetuses in Egypt [66].

In this study, we analyzed the effects of sex and breeding locations on the seroprevalence of *T. gondii* and *N. caninum* infection. The breeding location was identified as a risk factor for *T. gondii* where 15 May city had a significantly higher seroprevalence than the two other places. For *N. caninum*, both sex and location significantly affected the prevalence rate; females had a higher rate than males, and meanwhile, 15 May and Manshiyat Naser had higher rates than Ezbet El Nakhl. Notably, 15 May city recorded the highest prevalence for both *T. gondii* and *N. caninum*. This might be related to the more recent establishment of 15 May city, which has a larger desert area compared to the other two locations, and thus a higher chance of the presence of wild canines and felines. Another reason might be its distant location from the crowded areas of Cairo, which rendered it a main place for garbage collection. However, because this is the first study that investigated such factors in Egypt, other investigations are required to confirm these results and reveal the cause of variations. Meanwhile, numerous studies have found that location is a risk factor for the seroprevalence of *T. gondii* in pigs or other animals [21]. Concerning *N. caninum* in pigs, the available data are very scanty, both in Egypt and globally. Nevertheless, the location is considered a potential risk factor for *N. caninum* in other animal species [45,50,67,68]. Similarly, and taking into account the limited data concerning pigs, sex is regarded as a risk factor for *T. gondii* and *N. caninum* in cases of other animal species [50,69]. Age is usually also a risk factor for seropositivity to *T. gondii* or *N. caninum*, but this could not be assessed in our study, as all pigs were slaughtered at approx. 5 months old.

*Trichinella* spp. transmission primarily occurs in poor areas where veterinary services are unavailable, as well as in traditional small-scale “backyard” rearing of pigs for household and local use, particularly by the feeding of food waste [70,71]. *Trichinella* infection in pigs can be prevented through management practices such as grain feeding or cooking the feed garbage, indoor production, and removing rat carcasses from the premises [72,73,74,75].

In Egypt, larvae of *Trichinella* spp. have infrequently been found by trichinoscopy and/or digestion methods (Table 6). Interestingly, data on 33,812 slaughtered pigs were collected through a standard form from the El-Bassatin abattoir archives during one year of screening, and 359 (1.06%) were found to be infected [35]. However, Mohammed et al. [35] additionally tested a subsample of pigs with the digestion method and found a higher prevalence of 3.3%. Using artificial digestion, a prevalence rate of 13.3% was determined in rodents from Alexandria, and high infection rates were identified in rodents taken from and surrounding abattoirs [36]. Based on the isoenzyme electrophoretic patterns, *Trichinella spiralis* was identified in two stray dogs [37], two isolates from domestic pigs, and one isolate from a domestic dog from Egypt [34], and by PCR in domestic pigs [35]. Furthermore, human trichinellosis linked to the consumption of pork has been reported in Egypt [35,40,76,77,78].

Although current serodiagnostic approaches cannot replace traditional meat inspection for this zoonotic nematode, ELISA can be useful for surveillance programs and generally has an increased sensitivity compared to direct detection of larvae [79]. ELISA testing of 14,579 sera from pigs in 21 studies indicated an overall prevalence of 4.3% worldwide [75]. The seroprevalence rate obtained for *Trichinella* infection in pigs in our study was quite low (1.2%) and very similar to the infection rates determined by trichinoscopy at slaughter. Experimental studies have shown that *Trichinella* species have a different larval burden and immunological response in pigs at 30–70 days post-infection that can last up to 40 weeks [80,81,82,83]. Thus, the low seroprevalence of *Trichinella* spp. might be related to the younger age of tested pigs that are slaughtered at an age of 5 months. Likewise, the seroprevalence of *Trichinella* spp. in Vietnam increased with the pigs’ age: it was 1.1%, 1.2%, 32.9%, and 55.6% for pigs < 2 months, 2–8 months, 9–36 months, and >36 months, respectively [84]. Larger studies including different age groups and combining serological and direct methods should be envisaged to determine the infection rate in Egyptian pigs.

In conclusion, this study revealed variable seropositive rates against all tested parasites in pigs from Cairo, Egypt. Seropositivity was highest in *T. gondii* (45.8%) followed by *N. caninum* (28%) and *Trichinella* spp. (1.2%). Likewise, a high rate of mixed infections (18.7%) was noticed for *T. gondii* and *N. caninum.* Location was reported as a risk factor for seropositivity for *T. gondii,* while location and sex significantly affected *N. caninum* seropositivity in pigs. Future studies, including larger sample sizes and more diverse regions of Egypt, are required to highlight the national prevalence of these parasites and determine their potential public health risk. Furthermore, using molecular methods for genotyping and species identification of *T. gondii* and *Trichinella* spp., respectively, is critical to evaluate pigs’ role in the transmission of these parasites to humans in Egypt.

## Figures and Tables

**Figure 1 vetsci-10-00675-f001:**
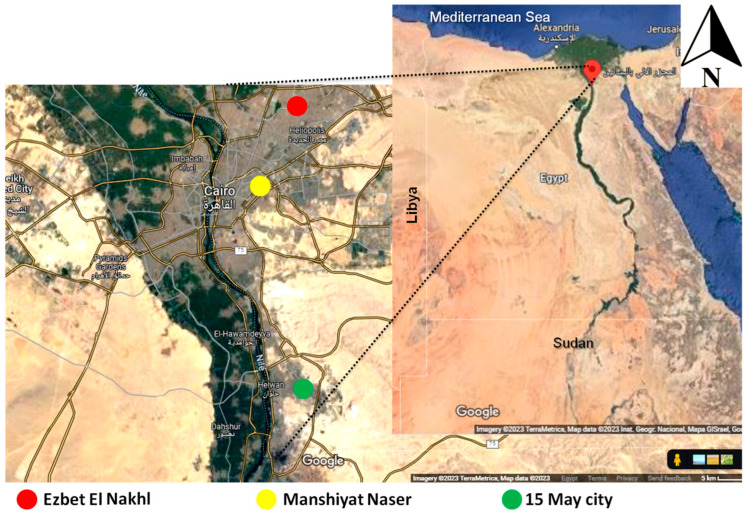
Geographical locations of sample origins. Serum samples were collected from pigs during slaughtering in the El-Bassatin abattoir, Cairo. These animals originated from three main slums of pig raising in three parts of Cairo, the capital governorate of Egypt: Ezbet El Nakhl (eastern Cairo), Manshiyat Naser (western Cairo), and 15 May city (southern Cairo).

**Table 1 vetsci-10-00675-t001:** Commercially available ELISA test kits used for detecting antibodies against *T. gondii*, *N. caninum*, and *Trichinella* species ^#^.

Infectious Agent	ELISA Test Kit ^$^	Antigen	Serum Dilution	Conjugate	Sensitivity *	Specificity *
*Toxoplasma gondii*	ID Screen^®^ Toxoplasmosis Indirect Multispecies	P30 antigen	1:10	Anti-multi-species IgG-HRP	98.36% (95% CI: 95.29–99.44)	99.42% (95% CI: 98.8–100)
*Neospora caninum*	ID Screen^®^ *Neospora caninum* competition Multispecies	Purified extract of *Neospora caninum*	1:2	Anti-*N. caninum*-HRP (detects IgG or IgM)	100% (95% CI: 98.8–100)	100% (95% CI: 99.63–100)
*Trichinella* species	ID Screen^®^ *Trichinella* Indirect Multispecies	*Trichinella* E/S antigen	1:20	Anti-multi-species-HRP (detects IgG or IgM)	90.7% (95% CI: 89.1–92.4)	100% (95% CI: 98.95–100)

^$^ All ELISA kits were manufactured by ID.vet Innovative Diagnostics, Grabels, France. * The sensitivity and specificity of the diagnostic kits were provided by the manufacturer of the kits. ^#^ detects antibodies to *T. spiralis*, *T. pseudospiralis*, *T. britovi* and *T*. *nativa*.

**Table 2 vetsci-10-00675-t002:** Seroprevalence of *T. gondii*, *N. caninum*, *Trichinella* spp. and mixed infections in pigs from Egypt.

Type of Infection	No. Tested	No. Negative (%)	No. Doubtful (%)	No. Positive (%)	95% CI *
*T. gondii*	332	152 (45.8)	28 (8.4)	152 (45.8)	40.4–51.3
*N. caninum*	332	217 (65.4)	22 (6.6)	93 (28.0)	23.3–33.2
*Trichinella* spp.	332	327 (98.5)	1 (0.3)	4 (1.2)	0.4–3.3
*T. gondii + N. caninum*	332	251 (75.6)	19 (5.7)	62 (18.7)	14.7–23.4

* 95% CI calculated according to the method described by http://vassarstats.net/, accessed on 22 July 2023.

**Table 3 vetsci-10-00675-t003:** Risk factors for *T. gondii* antibodies in pigs, Cairo, Egypt.

Analyzed Factor	No. Tested	No. Negative (%)	No. Positive (%)	OR (95% CI) *	*p-*Value ^#^
Sex					
Female	200	105 (52.5)	95 (47.5)	Ref.	Ref.
Male	132	75 (56.8)	57 (43.2)	0.8 (0.5–1.3)	0.499
Breeding place					
Ezbet El Nakhl	184	119 (64.7)	65 (35.3)	3.8 (1.8–7.9)	0.0003
Manshiyat Naser	108	48 (44.4)	60 (55.6)	0.6 (0.3–1.3)	0.259
15 May city	40	13 (32.5)	27 (67.5)	Ref.	Ref.

* Odds ratio and 95% confidence interval calculated by http://vassarstats.net/ (accessed on 22 July 2023). ^#^ *p* value was evaluated by Fisher’s exact test using online statistics software http://vassarstats.net/ (accessed on 22 July 2023) and GraphPad Prism version 5. Ref. is the value used as a reference.

**Table 4 vetsci-10-00675-t004:** Risk factors for *N. caninum* antibodies in pigs, Cairo, Egypt.

Analyzed Factor	No. of Tested	No. of Negative (%)	No. of Positive (%)	OR (95% CI) *	*p-*Value ^#^
Sex					
Female	200	124 (62)	76 (38)	Ref	Ref
Male	132	115 (87.1)	17 (12.9)	4.1 (2.3–7.4)	<0.0001
Breeding place					
Ezbet El Nakhl	184	160 (87)	24 (13)	Ref.	Ref.
Manshiyat Naser	108	50 (46.3)	58 (53.7)	7.7 (4.4–13.7)	<0.0001
15 May city	40	29 (72.5)	11 (27.5)	2.5 (1.1–5.7)	0.031

* Odds ratio and 95% confidence interval calculated by http://vassarstats.net/ (accessed on 22 July 2023). ^#^ *p* value was evaluated by Fisher’s exact test using online statistics software http://vassarstats.net/ (accessed on 22 July 2023) and GraphPad Prism version 5. Ref. is the value used as a reference.

**Table 5 vetsci-10-00675-t005:** Seroprevalence of *N. caninum* antibodies in animals in Egypt.

Animals	Location	Study Year	No. Tested	No. Positive (%)	Diagnostic Method	History	Reference
Buffaloes	Cairo slaughterhouse	1995	75	51 (68)	NAT	NS	[58]
Camels	Cairo slaughterhouse	1995	161	6 (3.7)	NAT	NS	[59]
Cattle	Sharkia	NS	93	19 (20.43)	ELISA	NS	[60]
Cattle	Qena and Sohag	2014–2015	301	57 (18.9)	ELISA	NS	[61]
Cattle	Fayoum, Giza, Beni-Swief, and Menia	2017	100	29 (29) ^1^	ELISA	Infertility and abortion	[51]
Camels	Red Sea, Qalyubia, Kafr, and El Sheikh	2018–2019	282	31 (11)	ELISA	150 camels had a history of abortion	[53]
Cattle	Menoufia	2017–2018	262	32 (12.21) ^1^ 39 (14.89) ^2^ 3 (1.5) ^3^	ELISA	NS	[62]
Buffaloes			244	17 (6.97) ^1^ 33(13.52) ^2^ 12 (4.92) ^3^			
Sheep	Alexandria, Gharbia, Menofia, and Qalyubia	2017–2018	430	37 (8.6)	ELISA	NS	[63]
Cattle	Giza, El Fayoum, and Beni Suef		116	35 (30.17)	ELISA	Aborted bovine fetuses	[52]
Camels	Aswan	2018–2021	460	18 (3.9)	ELISA	NS	[44]
Sheep and Goats	Dakahlia, Beni Suef, Qena, and Red Sea	2016–2021	239 121	37 (15.5) 6 (5)	ELISA	NS	[45]
Dogs and Cats	Variable	2019–2022	172 51	10 (5.8) 2 (3.9)	ELISA	-	[49]
Cattle	Beheira		358	88 (24.6)	ELISA	NS	[50]

^1^ IgM-ELISA positive, ^2^ IgG-ELISA positive, ^3^ mixed IgM, IgG-ELISA positive. Abbreviations: ELISA, enzyme-linked immunosorbent assay; NAT, neospora agglutination test; NS, not stated.

**Table 6 vetsci-10-00675-t006:** Prevalence of *Trichinella* sp. larvae in pigs in Egypt.

Sample Types	Governorates	Study Year	No. Tested	No. Positive (%)	Diagnostic Methods	Species	Reference
pork sausage	Alexandria	NS	100	6 (6) 4 (4)	Trichinoscope Digestion method	*T. spiralis*	[38]
abattoir meat	Assiut and Sohag	2006–2007	150	6 (4) 5 (3.33)	Trichinoscope Digestion methods	*T. spiralis*	[40]
abattoir meat	El-Minia	2014–2015	100	0	Muscle squash and digestion methods	-	[19]
abattoir meat	Cairo	2018–2019	184	2 (1.08)	Trichinoscopic and histopathological examination	*T. spiralis*	[41]
abattoir inspection	Cairo	2020	33,812 *	359 (1.06)	Trichinoscopic examination	*Trichinella* sp.	[35]
abattoir meat	Cairo	2020	170	6 (3.35) 4 (2.35)	Digestion methods Trichinoscope	*T. spiralis* ^#^	[35]

* Data were collected through a standard form from the Al-Basateen abattoir archives. ^#^ *Trichinella spiralis* was identified based on PCR of the GAPDH gene.

## Data Availability

All data generated and analyzed during this study are included in this published article. The raw data supporting the findings of this study are available from the corresponding author on request.

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
