# Peer review of "Seroprevalence of Toxoplasma gondii, Neospora caninum and Trichinella spp. in Pigs from Cairo, Egypt"

_vetsci, 2023, doi:10.3390/vetsci10120675_

Round 1

Reviewer 1 Report

Comments and Suggestions for Authors

In this work, the authors investigated the seroprevalence of Toxoplasma gondii, Neospora caninum and Trichinella spp. in pigs from Cairo, Egypt. The results were well presented and discussed. Giving a review of similar studies in the discussion section enriched this work.

I only suggest an English review to correct minor things.

Comments on the Quality of English Language

An English review to correct minor things is advisable 

Author Response

Reviewer-1

Comments and Suggestions for Authors:

In this work, the authors investigated the seroprevalence of Toxoplasma gondii, Neospora caninum and Trichinella spp. in pigs from Cairo, Egypt. The results were well presented and discussed. Giving a review of similar studies in the discussion section enriched this work.

Authors’ response

We are greatly thankful to reviewer #1 for the encouraging comments

I only suggest an English review to correct minor things.

Authors’ response

The manuscript text was subjected to English editing for the structure and grammatical mistakes and ensure it is understandable and mostly reads well.

All revised parts were indicated in blue colored fonts.

Reviewer 2 Report

Comments and Suggestions for Authors

The article is well written, the data are properly presented and explained. However, there are several major issues that require attention before the manuscript can be accepted. 

This manuscript provides an interesting contribution to prevalence of Toxoplasma gondii, Neospora caninum and Trichinella spp. in pigs from Egypt. Risk factors for the diseases have also been considered.

I believe that the manuscript can be enriched if the following suggestions are considered:

 1. L173-L175. Provide details of the statistical test for calculating odds ratios.

 2. Table 2. If in doubt about the positivity or negativity of the test, confirm with another test.

 3. Table 3. P-value was significant for sex and breeding place, but the odds ratios are not greater than 1, so they do not have an influence on the dependent variable.

 4. L276-L282. The discussion on breeding place and T. gondii has to be rewritten as it has no effect on prevalence.

Comments on the Quality of English Language

Minor editing of English language required

Author Response

Reviewer-2

Comments and Suggestions for Authors:

The article is well written, the data are properly presented and explained. However, there are several major issues that require attention before the manuscript can be accepted.

This manuscript provides an interesting contribution to prevalence of Toxoplasma gondii, Neospora caninum and Trichinella spp. in pigs from Egypt. Risk factors for the diseases have also been considered.

Authors’ response

We are greatly thankful to reviewer #2 for the encouraging comments.

I believe that the manuscript can be enriched if the following suggestions are considered:

  1. L173-L175. Provide details of the statistical test for calculating odds ratios.

Authors’ response

Description of used methods was illustrated in text as the follows:

“From this website, calculator 3 was selected from the clinical research calculators. Then in squares of data entry, the numbers of negative (absent) and positive (present) samples from group 1 (regarded as reference group) were added, respectively. Similarly in lower squares, numbers of negative and positive samples were added as in previous. In these tests, we used the two-tailed P values of Fisher exact test” (Lines 207-211)

.2. Table 2. If in doubt about the positivity or negativity of the test, confirm with another test.

Authors’ response

According to the manufacturer instructions, results are judged as negative, doubtful and positive based on % of the inhibition. As a confirmatory step, the samples values occurred between the doubtful ranges were tested again and only samples that gave similar values were considered doubtful. We followed this method in many of our previously published reports

DOI: 10.3389/fvets.2023.1122092

DOI: 10.3390/pathogens11121464

DOI: 10.3390/ani12233327

DOI: 10.3389/fcimb.2022.1042279

  1. Table 3. P-value was significant for sex and breeding place, but the odds ratios are not greater than 1, so they do not have an influence on the dependent variable.

Authors’ response

We apologize for this mistake; the value was corrected after confirmation with two statistical methods vassarstats.com and GraphPad Prism. In addition, all values were revised again to confirm the correctness and accuracy.

  1. L276-L282. The discussion on breeding place and T. gondii has to be rewritten as it has no effect on prevalence.

Authors’ response

We apologize for this mistake; the value of odd ratio was corrected after confirmation with two statistical methods vassarstats.com and GraphPad Prism. In addition, all values were revised again to confirm the correctness and accuracy.

The sentence was rewritten (lines: 325-329) as follow “In this study, we analyzed the effects of sex and breeding locations on the seroprevalence of T. gondii and N. caninum infection. Pigs breeding location was identified as a risk factor for T. gondii where 15 May city had a significantly higher seroprevalence than the two other places”.

Comments on the Quality of English Language

Minor editing of English language required

Authors’ response

The manuscript text was subjected to English editing for the structure and grammatical mistakes and make sure it is understandable and mostly reads well.

All revised parts were indicated in blue colored fonts.

Reviewer 3 Report

Comments and Suggestions for Authors

Congratulation for the article, this provides information on parasitic diseases, some of them zoonotic, in pigs.

However, it is necessary to improve some sections for a better understanding.

- Introduction: How many cases of trichinellosis/toxoplasmosis are describe in human beings from Egypt?

- Material and methods. Could you provide more information about where the animals are housed, biosecurity measures (e.g., presence of rodents), type of diet, and the differences between some geographic areas and others? What is the origin of pigs? 

Are these pigs housed with other slaughter animals for human consumption such as lamb? If this is so, it should be taken into account and included in the discussion

- Neosporosis is an important parasitic disease in ruminants because it causes abortions. However, the discussion does not explain in depth the significance of the Neospora caninum seroprevalence data. It seems that it is an additional data of information since the ELISA technique has been performed, but it is not clearly justified.

Author Response

Reviewer-3

Comments and Suggestions for Authors:

Congratulation for the article, this provides information on parasitic diseases, some of them zoonotic, in pigs.

However, it is necessary to improve some sections for a better understanding.

Authors’ response

We are greatly thankful to reviewer #3 for the encouraging comments.

-Introduction: How many cases of trichinellosis/toxoplasmosis are describe in human beings from Egypt?

Authors’ response

Many cases of toxoplasmosis were reported in Egypt and reviewed by Abbas et al., 2020. We added the following sentences “Many cases of human toxoplasmosis were reported in Egypt and reviewed previously. The seroprevalence of T. gondii in the Egyptian human population ranged from 2.5 to 67.4%, and toxoplasmosis is commonly regarded by Egyptian physicians as a cause of abortions and complications in pregnant women; nonetheless, published studies are not well-structured, and there is a lack of definitive diagnosis [20-22].” (lines 94-99)

For human trichnellosis, we added the following sentence “Human trichinellosis linked to the consumption of pork has recently been reported from Egypt based on ELISA, the seroprevalence rate of T. spiralis (IgG) was 10% (9/90) [35] and 60.9% (56/92) [40] in the examined humans.” (lines 124-126)

-Material and methods. Could you provide more information about where the animals are housed, biosecurity measures (e.g., presence of rodents), type of diet, and the differences between some geographic areas and others? What is the origin of pigs?

Authors’ response

We revised the related information in the text to respond the required new information. (lines 137-151)

 “All pigs were raised in backyard pens containing 50 to 100 pigs in the slums of Cairo governorate by garbage collectors as one of their main sources of income. These rearing pens and slums are lacking many procedures of sanitary conditions or biosecurity and fed mainly on garbage. This condition allowed the high contact with free roaming animals such as dogs, cats, rodents, birds increasing the incidence with infection with many biohazard agents. In addition, some livestock animals especially sheep are also reared with pigs. The majority of pigs reared for fattening and slaughtered at the age of 5 months. They are typically sold to pig merchants who collect animals from different small-scale farmers for shipping them to the abattoir. Pigs originated from the following slums: Manshiyat Naser, Cairo's largest pig farming slum in the western region (30.0362°N, 31.2783°E) (n = 108; female 75, male 33); Ezbet El Nakhl (30.1393°N, 31.3244°E) (n = 184; female 125, male 59) in eastern Cairo region; and 15 May city (29.8579°N, 31.3885°E) (n = 40; all of them are male) in southern Cairo region (Figure 1). The pigs from various herds are held together in abattoir pens before slaughtering without identification.”

Are these pigs housed with other slaughter animals for human consumption such as lamb? If this is so, it should be taken into account and included in the discussion

Authors’ response

This information was added in materials and methods

“These rearing pens and slums are lacking many procedures of sanitary conditions or biosecurity and fed mainly on garbage. This condition allowed the high contact with free roaming animals such as dogs, cats, rodents, birds increasing the incidence with infection with many biohazard agents. In addition, some livestock animals especially sheep are also reared with pigs.” (lines 139-142)

In discussion we added these new statements;

“Although N. caninum might not be a serious parasite for pig industry, its detection in pigs might constitute a major risk for ruminants and other animals because of mixed breeding in the same place.” (lines 293-295)

Also, discussion already contained relevant information. (lines 278-290)

-Neosporosis is an important parasitic disease in ruminants because it causes abortions. However, the discussion does not explain in depth the significance of the Neospora caninum seroprevalence data. It seems that it is an additional data of information since the ELISA technique has been performed, but it is not clearly justified.

Authors’ response

We added (lines 283-290) the following paragraph “Nonetheless, serum samples from aborted cows revealed N. caninum infections in addition to Coxiella burnetii [51] or Bovine Viral Diarrhoea Virus [52]. Furthermore, the number of N. caninum seropositive camels increased mostly in females with an abortion history [57]. On the other hand, the pathogenesis of neosporosis and its consequences in pigs remain unclear [28-29]. Neospora caninum experimental infection in pigs can be transferred through the placenta and produce reproductive problems such as mummified foetuses, particularly in the first and second gestational thirds [28].”

Reviewer 4 Report

Comments and Suggestions for Authors

Dear authors,

The manuscript title is “Seroprevalence of Toxoplasma gondii, Neospora caninum and Trichinella spp. in pigs from Cairo, Egypt” and it aims to assess the antibodies seroprevalence of these parasites in pigs from Cairo, Egypt. The topic falls within the aims and scope of the journal. The limitation I see in this study is that it has been done in only one abbatoir and not so many samples, so it seems more a regional study rather than a national one. On the other hand, it is important to have the first report about Neospora in the country. English should be improved, at least in abstract and introduction please.

Some particular suggestions/comments will be done here:

- Lines 38/178/Table 2/222 and wherever in the manuscript – please add a decimal place to 28% (28.0%?) in order to be coherent with the other prevalence

- Line 47 – I suggest not to repeat in keywords, words that are already in the title

- Lines 53/54 – you should transform this site in a bibliographic reference here

- In Introduction, I suggest you to add information about the impact of these parasites in animal/human health, I miss mentions about symptoms/clinical signs. What do these parasites cause to animals and humans?

- Line 119 – You need to add how/from where exactly you collected blood samples, please

- Line 219 - : instead of ; please

- Line 234 – You can not refer to a high infection rate as you assessed antibodies and not Toxoplasma itself.

- Line 251 – “higher”

- Line 306 – is it 1.06%? Is the % missing?

- Line 319/323 – I suggest you to discuss also Trichinella results taking into account the early age of the slaughter pigs as antibodies of this parasite may need 2 months to reach a peak after infection

I suggest you to add in conclusions some limitations about your study, namely the small number of samples, the regional character, the diagnostic methodology used and others you may have.

Author Response

Reviewer-4

Comments and Suggestions for Authors:

The manuscript title is “Seroprevalence of Toxoplasma gondii, Neospora caninum and Trichinella spp. in pigs from Cairo, Egypt” and it aims to assess the antibodies seroprevalence of these parasites in pigs from Cairo, Egypt. The topic falls within the aims and scope of the journal.

Authors’ response

We are greatly thankful to reviewer #4 for the encouraging comments.

The limitation I see in this study is that it has been done in only one abbatoir and not so many samples, so it seems more a regional study rather than a national one. On the other hand, it is important to have the first report about Neospora in the country.

Authors’ response

Swine/pigs stock number in Egypt is very low (about 11,000 heads) and Christians or tourists are the main consumers. El-Bassatin abattoir is largest in Egypt and the only licensed place for pig slaughtering in Egypt. Thus, pork meat and products from slaughtered pigs in El-Bassatin abattoir are consumed by Christians and tourists in many governorates other than Cairo, the capital of Egypt.  We already highlighted the novelty of point of seroprevalence of N. caninum in pigs in Egypt in many occasions in the text (lines 46, 62, 128)

English should be improved, at least in abstract and introduction please.

Authors’ response

The manuscript text was subjected to English editing for the structure and grammatical mistakes and make sure it is understandable and mostly reads well.

All revised parts were indicated in blue colored fonts.

Some particular suggestions/comments will be done here:

- Lines 38/178/Table 2/222 and wherever in the manuscript – please add a decimal place to 28% (28.0%?) in order to be coherent with the other prevalence

Authors’ response

Corrected as suggested, (lines 42, 55, 216) and table 2.

- Line 47 – I suggest not to repeat in keywords, words that are already in the title

Authors’ response

Keywords were corrected as follow “Pigs, Egypt, toxoplasmosis, neosporosis, trichinellosis, ELISA” (lines 64)

- Lines 53/54 – you should transform this site in a bibliographic reference here

Authors’ response

Corrected as suggested Reference 2. (line 70)

- In Introduction, I suggest you to add information about the impact of these parasites in animal/human health, I miss mentions about symptoms/clinical signs. What do these parasites cause to animals and humans?

Authors’ response

We added the following sentences

For Toxoplasmosis “Infections in pigs are usually asymptomatic or with non-pathognomonic symptoms, and severe clinical toxoplasmosis is rare [11]. Human primary infection with T. gondii normally manifests as mild flu-like symptoms in immune-competent hosts, however it can induce life-threatening infections in immune-compromised persons. Furthermore, T. gondii transplacental transmission during pregnancy might result in significant complications such as abortion, stillbirth, or newborn anomalies [8]”. (lines 82-87)

“Many cases of human toxoplasmosis were reported in Egypt and reviewed previously. The seroprevalence of T. gondii in the Egyptian human population ranged from 2.5 to 67.4%, and toxoplasmosis is commonly regarded by Egyptian physicians as a cause of abortions and complications in pregnant women; nonetheless, published studies are not well-structured, and there is a lack of definitive diagnosis [20-22].” (lines 94-99)

For Neosporosis “Abortions and newborn mortality are serious implication of neosporosis in livestock, particularly in cattle [23-24]” (lines 105-106).

For Trichinellosis “Pork was the most common source of human trichinellosis, which can cause clinical symptoms such as myalgia, diarrhea, fever, facial edema, headaches, and may even be fatal” (lines 115-117)

- Line 119 – You need to add how/from where exactly you collected blood samples, please

Authors’ response

We added this information

“Blood was collected in plain tubes without anti-coagulants directly from the slaughtered animals during bleeding according to El-Bassatin abattoir regulations for pig slaughtering and pork processing. All samples were labeled and transferred on ice to the Reference Laboratory at the Animal Health Research Institute (AHRI), Giza. Upon arrival to the lab, blood samples were kept overnight for serum separation then, centrifuged at 900 × g for 10 minutes. Serum samples were aliquoted and kept at – 20 oC until being used. Aliquots from these samples were sent on ice to the Faculty of Veterinary Medicine, South Valley University, Qena for ELISA testing” (lines 152-159).

- Line 219 - : instead of ; please

Authors’ response

Corrected as suggested. (line 258)

- Line 234 – You can not refer to a high infection rate as you assessed antibodies and not Toxoplasma itself.

Authors’ response

We corrected the sentence as follow “Our detected seroprevalence of 45.8% for T. gondii in the assessed backyard pigs indicates a potential risk to the health of pig fatteners, abattoir workers, and consumers” (lines 272-274)

- Line 251 – “higher”

Authors’ response

Corrected as suggested. (line 300)

- Line 306 – is it 1.06%? Is the % missing?

Authors’ response

Corrected (1.06%). (line 355)

- Line 319/323 – I suggest you to discuss also Trichinella results taking into account the early age of the slaughter pigs as antibodies of this parasite may need 2 months to reach a peak after infection

Authors’ response

We added the following sentences “Experimental studies have shown that Trichinella species have a different larval burden and immunological response in pigs at 30-70 days post-infection that can last up to 40 weeks [80-83]. Thus, the low seroprevalence of Trichinella spp. might be related to the younger age of tested pigs that are slaughtered at age of 5 months. Likewise, the seroprevalence of Trichinella spp. in Vietnam increased with pigs age: it was 1.1%, 1.2%, 32.9%, and 55.6% for pigs < 2 months, 2 - 8 months, 9 - 36 months, and > 36 months, respectively [84]. Larger studies including different age groups and combining serological and direct methods should be envisaged to determine the infection rate in Egyptian pigs.” (lines 370-378).

I suggest you to add in conclusions some limitations about your study, namely the small number of samples, the regional character, the diagnostic methodology used and others you may have.

Authors’ response

We added the following sentences “Future studies including large sample sizes and diverse regions of Egypt are required to highlight the national prevalence of these parasites and determine the potential public health risk. Furthermore, using molecular methods for genotyping and species identification of T. gondii and Trichinella spp., respectively, is critical to evaluate pigs' role in transmission to humans in Egypt.” (lines 388-392)

Round 2

Reviewer 2 Report

Comments and Suggestions for Authors

The authors have responded to each of the reviewers´ comments and suggestions.

Reviewer 3 Report

Comments and Suggestions for Authors

Thank you for your comments, I think the document have improved and It could be accepted in the present form.